# A Metric Driven Approach to Mixed Precision Training

Gil Tabak*, Mitchelle Rasquinha*,

*Google. *tabakg@google.com, mrasquinha@google.com*

*Abstract*—As deep learning methodologies have developed, it has been generally agreed that increasing neural network size improves model quality. However, this is at the expense of memory and compute requirements, which also need to be increased. Various efficiency techniques have been proposed to rein in hardware costs, one being the use of low precision numerics. Recent accelerators have introduced several different 8-bit data types to help accommodate DNNs in terms of numerics. In this paper, we identify a metric driven methodology to aid in the choice of numerics. We demonstrate how such a methodology can help scale training of a language representation model. The technique can be generalized to other model architectures.

## I. Introduction

The wide success of Deep neural networks has led to continued increases in model sizes and the computing resources needed to train them. Further the introduction of Large Language Models has dramatically increased this demand for training and serving. Such a massive demand for system resources outperforms Moore's Law and hardware capabilities by a wide margin. Several model efficiency techniques have been proposed to mitigate this unprecedented demand [5], [11], including the use of reduced precision operations. Quantization - the process of reducing the number of bits used to represent a number, can improve the performance of deep learning models by reducing the amount of memory and computational power required.

Deep learning training today includes a wide range of data types. Common floating point types include IEEE single precision, FP32 mode for single precision, IEEE half precision [8], and bfloat16 [1], [7]. More recently 8 bit types have been introduced in deep learning accelerators with trade-offs between the exponent and the mantissa bits to accommodate the needs of different operations within a model. In addition to floating bit representations integer hardware has also been introduced and has two key advantages – (1) Area and energy efficient hardware units (2) Fewer sources of introduced error within the accumulation hardware unit. A given neural network accelerator may provide a few different numerical data types depending on its applicability to operations within the model structure. While the choice of data types provides flexibility in training, it is often a complex search for the right set of numerics for a given model. At bit widths of 16 bits and lower a careful quantization application is required, without which model quality suffers.

We make the following contributions
- Develop a metric driven methodology to guide the use of different low precision numeric formats.

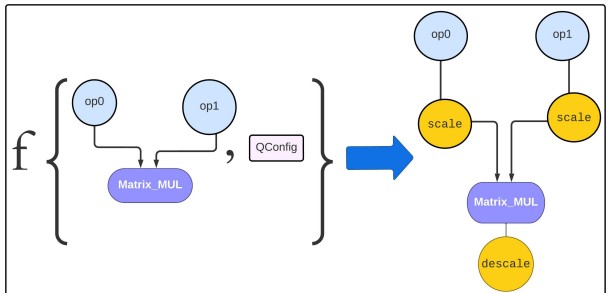

Fig. 1. Illustration of the neural network graph modification during quantization.

- Demonstrate the methodology predicts training quality using different mixed precisions for the BERT model.

## II. Related Work

The use of low precision numerics in inference has been widely studied and as shown significant benefits in terms of compressing the models while retaining model quality. The use of 8 bit integer for inference was introduced in [6]. A comprehensive list of different techniques to use low precision numerics can be found in [3]. Recently, accelerators have introduced multiple low precision formats [9], [13], [14] further extending their use in both training and serving workloads. [16] have shown that 8-bit floating representation can be used to train convolutional neural networks, with the help of stochastic rounding.

FP8 and Int8 hardware implementations feature reduced bit width multiply-accumulate(MAC) units, thus attaining very high energy, latency, and bandwidth gains compared to 32 and 16-bit counterparts. More aggressive bit width reductions, also known as binary quantization have also been explored in [12]. In this paper we focus on the use of 8 bit low precision formats for training neural networks.

## III. Methodology

Quantization is typically applied to compute intensive matrix multiplication operations within a neural network. We study uniform integer quantization with dynamic scaling for improved model performance and power. For each operand of the dot product, quantization is described as follows:

$$\bar{X}_{\text{int8}} = \lceil \frac{X_{\text{bfloat16}}}{\delta} \rfloor, \ \delta = \frac{\max(|X_{\text{bfloat16}}|)}{2^{n-1} - 1} \qquad (1)$$

$X_{\text{bfloat16}}$ is the high precision floating point format and $\bar{X}$ is the quantized counterpart, $\delta$ is the quantization step size and $n$ is the bit width of the quantized tensor. The quantization step size is calculated for each quantized operand at every training step commonly referred to as dynamic quantization. There are several choices for the rounding function $\lceil . \rfloor$ with the default IEEE rounding technique being round to the nearest even (RTNE). We study three different 8-bit formats, namely INT8, E4M3 and E5M2. E4M3 and E5M2 are jointly referred to as FP8 formats. FP8 quantized values conform to the rules described in [9]. An FP8 format shares the 8 available bits between exponent$(e)$ and mantissa$(m)$ and can be more generally described as an $EeMm$ format. One bit is reserved for the sign. It is common for the exponent itself to be biased to shift the expressible range.

The framework level quantization can vary between implementations and the following are specific to our implementation:

- In all 8-bit formats, out-of-range tensor elements are based on the following rule: Tensor elements over the max expressible value are saturated at the max expressible value. Tensor elements whose absolute value is smaller than the smallest sub-normal are represented by zero.
- We use two different forms of rounding - (1) round-to-nearest-even and (2) stochastic rounding
- In both FP8 and INT8, some form of scaling is applied before and after the matrix multiplication. After matrix multiplication, descaling is applied to the output as shown in figure 1. The output prior to re-scaling is typically not represented by 8-bits and depends on the multiple-accumulate unit.
- The scaling may be done at the *tensor level* meaning a single number is used for each scaling, or at finer levels of granularity on non-contracting dimensions. We present results for a variation of *scale granularity*.
- Throughout, we assume scaling is always done to align the absolute max value to the maximum expressible value of the chosen format with *symmetric quantization*. The maximum expressible values are 448 for E4M3, 57344 for E5M2 and 127 for Int8.

The use of reduced precision numerics is a lossy compression technique. The choice of quantization parameters plays a critical role in determining the magnitude of the introduced error.

### A. Model evaluation

Deep learning architectures comprise several computations that can be abstracted into a few basic operations such as convolution and matrix multiplication. Highly optimized compute kernels are available from different machine learning frameworks for these operations.

We evaluate our methodology on the BERT architecture from [2], [15]. The baseline model is trained in bfloat16 and tensor inputs to all the major matrix multiplications were sampled to compute the mean, variance, skew and Kurtosis. Figure2 plots the distributions for one such set of tensors.

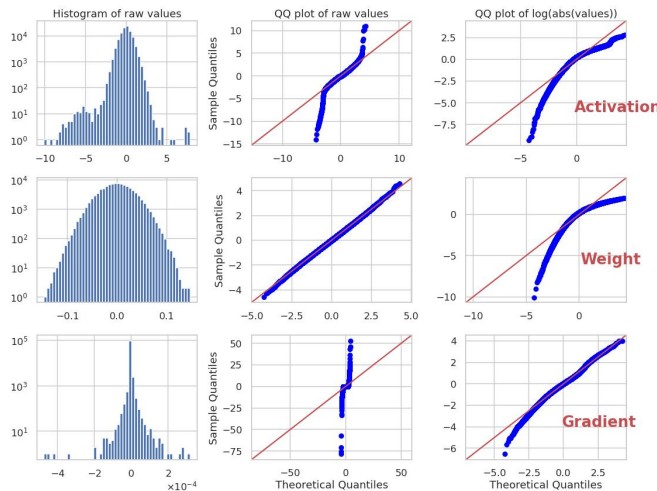

Fig. 2. Distributions of the input tensors to the query projection dot operation in the forward and backward passes. Red text denotes the tensor type.

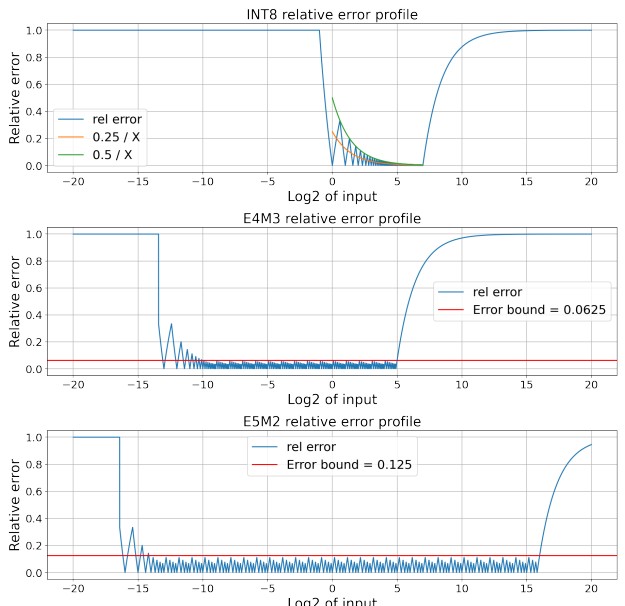

Fig. 3. A comparison of the relative error profile of INT8 and two FP8 formats, assuming RTNE.

In general we find that the weights have a normal distribution and the gradients have a log-normal distribution. The reference model is available at [10]

### IV. RESULTS

An essential component to minimizing overall model quality degradation is to minimize per operation quantization error. The quantization error depends on the distribution of the high precision tensors and the properties of the reduced-precision format. The quantization error can be categorized into (1) clipping error (2) rounding error. Clipping error, is the loss of accuracy due to values lying outside the dynamic range of a format i.e overflow or underflow. In our implementation all overflow values are capped at the max value and all

| | RHS (RTNE) | | | LHS (RTNE) | | | gradient (RTNE) | | | gradient (Stochastic) | | |
|---|---|---|---|---|---|---|---|---|---|---|---|---|
| | int8 | e4m3 | e5m2 | int8 | e4m3 | e5m2 | int8 | e4m3 | e5m2 | int8 | e4m3 | e5m2 |
| **tensor** | 17.14 | 17.12 | 17.13 | 17.01 | 17.11 | 17.11 | 1.4 | 17.07 | 17.15 | 12.73 | 17.12 | 17.13 |
| **channel** | 17.12 | 17.09 | 17.09 | 17.20 | 17.18 | 17.10 | 2.3 | x | x | 17.12 | x | x |
| **fine-grained** | 17.13 | 17.05 | 17.09 | 17.13 | 17.10 | 17.13 | 8.2 | x | x | 17.14 | x | x |

TABLE I
AREA-UNDER-THE-CURVE (AUC) FOR THE EVAL ACCURACY IN BERT TRAINING.

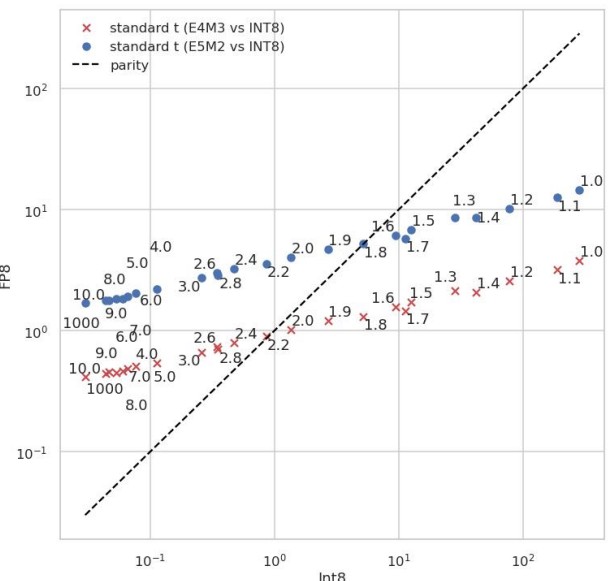

Fig. 4. Student's T-Distribution of the quantization error when INT8, E4M3 and E5M2 were each used as input data types.

underflow values are represented by zero. Rounding error is the loss of accuracy due to values lying between numbers (in the low-precision domain) and varies based on the rounding parameters. Figure 3 plots the ranges of the three different 8-bit formats, illustrating the trade-offs made between them. The relative error is defined as 2 for a given value $v$ where $\tilde{v}$ indicates the reduced-precision value.

$$RE = \frac{|v - \tilde{v}|}{v} \qquad (2)$$

*While INT8 captures values in a relatively narrow range to high precision, FP8 formats trade off high precision for a wider dynamic range.*

### A. Quantized matrix multiplication: An illustrative example

To demonstrate the differences between the precision of quantized matrix multiplication using FP8 versus INT8 among different input distributions we conducted a probabilistic error analysis. We chose the backward error based on the inner-product level definition given in [4]. Analysing the backward error clearly shows the differences between the quantization format of choice.

The error analysis assumes matrices of size $512 \times 512$ sampled using a t-distribution using a range of normality parameters (annotated in the plot). The backward error is given by 3 for inputs $L$ and $R$ where $\cdot$ indicates matrix multiplication and $Q(\cdot, \cdot)$ indicates quantized matrix multiplication (using per-vector scaling).

$$BE = \frac{|L \cdot R - Q(L, R)|}{|L| \cdot |R|} \qquad (3)$$

While the error can vary widely for INT8 depending on the heavy-tailedness of the inputs, it is much more constrained for FP8 formats. In our example we found E4M3 has a smaller error than E5M2, this may differ depending on the distribution and quantization methodology. For example, if the magnitude of the tensor entries varies widely for different inner-products, E5M2 will enable more flexibility even when using tensor-level quantization.

### B. BERT training results

To test the suitability of each format for different tensors, we varied a subset of the quantization parameters applied to a subset of the tensors. We broadly categorised tensors into RHS, LHS, and gradient categories. Gradients are always upstream gradients. The LHS were the activations, except inside the self-attention mechanism, where they refer to the key (in the keys times query computation) or probabilities (in the probability times value computation).

In Table I we show the area-under-the-curve (AUC) of the eval accuracy, to compare both converged and non-converged runs. An AUC of 17.13 was measured for the baseline. The standard deviations for experiments with converged runs were in the range of $[0.02, 0.08]$. While 'tensor' refers to using a single value for scaling the entire tensor, 'channel' refers to using a tensor for each non-batch/non-contracting dimension (batch dimension here refers to the matrix multiply batch). The 'fine-grained' level also includes batch dimensions, excepting the axis corresponding to individual examples.

As the RHS (mostly weights) were not heavy-tailed, the INT8 format was lossless regardless of the quantization granularity level. In comparison, the FP8 formats produced very close results, with degradation essentially within the noise level. Meanwhile, applying INT8 to the LHS (mostly activations, with a higher dynamic range than the weights) produced a slightly more noticeable degradation at the tensor level. However, this degradation can be overcome by using finer granularities. Finally, the gradients are extremely heavy-trailed. Using int8 without stochastic rounding never converged, although there was a clear pattern of improvement as we increased the level of granularity. Both FP8 formats converge when applied for gradients with RTNE. Finally, when considering stochastic rounding for upstream gradients, the

INT8 results still did not converge when applying tensor level quantization. It was necessary to use finer scales to achieve convergence. Both FP8 formats performed at the baseline level.

While the results provided are restricted to a single model, we believe the methodology is more widely applicable to other classes of models and can be evaluated on any ML accelerator with low precision numerics support. Additional framework support for applying the quantization technique is also required for an evaluation.

## V. CONCLUSION

We have identified a methodology to use different low precision numerical formats. At small bit widths of 8 bits and below, the minimal dynamic range requires careful mapping of operations within the model to the different multiply-accumulate units on the underlying hardware. This step is crucial to realizing the gains from low precision numerical formats without compromising the quality requirements of the model. The search space for bit width allocation increases exponentially with more layers and more numerical formats. Future work aims at identifying metrics that can help narrow the search space based on information within the baseline high precision tensors.

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
