# OpenReview forum: "A Metric Driven Approach to Mixed Precision Training"
_iscaconf.org/ISCA/2023/Workshop/ASSYST — ASSYST Oral_

### Official Review · Reviewer_2LFv · 2023-05-02
**Lacks clarity and Novelty**

**Rating:** 3
**Confidence:** 3

**Review:**

Overall, the paper lacks proper contribution. The introduction fails to summarize the challenges and relate the solution with the challenges. On top of that, the solution to selecting a proper bit-width for model is not unique to Bert or large-language model. It seems Bert is just a case study here. The paper needs further polishing and writing improvement.

**Review (Strengths/Weaknesses):**

First, the writing quality needs to improve. Overall, the challenges and their relation to large language model is not clear.
Second, in the vision domain, existing work already adopts mixed bit precision training by selecting and training the level of bit-width through gradient based updates. Hence, I do not find the contribution significant enough.

**Reviewer Expertise:**

Knowledgeable: I used to work in this area and/or I try to keep up with the literature but might not know the latest developments.

---

### Official Review · Reviewer_au4o · 2023-05-05
**Clear and readable paper, but possibly under-evaluated**

**Rating:** 5
**Confidence:** 2

**Review:**

The paper discusses three quantized formats: int8 and two fp8 formats, and claims to provide a methodology for determining which formats will perform best for which tensors. In the first half, the authors evaluate these three formats, showing relative error when representing different real numbers with each format and the error that they produce when multiplying matrices. In the second half of the paper, the authors discuss training BERT with these formats and show the final accuracy achieved.

Overall, the paper is reasonably clear and well-written and the problem is interesting. The authors also provide a qualitative explanation of how the input data formats impact the final accuracy of training, which they link back to Figure 2 showing the distributions of the input tensors. However, I wish that the paper were more fully developed to link the numerical results in the first half to the second---I feel like Section 4.2 is written to explain whatever (possibly random) variations arise in Table 1 instead of defining and evaluating a testable hypothesis.

**Review (Strengths/Weaknesses):**

# Strengths
- The paper provides reasonable results and an interesting description of the impact of different quantization types.
- The paper is clear about its claims and limitations, and seems to provide good motivation for future work.
# Weaknesses
- The authors only train one model.
- The authors only consider 8-bit data types---one integer type and two float types. It would be interesting to see a wider range of bit-widths, which might permit stronger conclusions.
- The authors do not show any error bars on the data in Table 1, which makes it hard to determine what is the result of noise and what is statistically significant.

**Reviewer Expertise:**

Little or no familiarity.

---

### Official Review · Reviewer_zSLk · 2023-05-05
**Review for A Metric Driven Approach to Mixed Precision Training**

**Rating:** 6
**Confidence:** 3

**Review:**

This paper presents a methodology to explore the impact of reduced-precision numbers on training machine learning models. They apply their methodology to the BERT model by quantizing tensors in the model, and then training models with quantized tensors at different 8-bit formats. The paper evaluates the error of quantized matrix multiplies as well as end-to-end-trained BERT with quantization for comparison.

**Review (Strengths/Weaknesses):**

### Strengths:
+ nice overview of how practitioners might approach quantizing portions of their own models
+ preliminary results demonstrate the benefits of quantization for a large model like BERT
+ I appreciated that the paper presented results on both individual tensor operations and end-to-end results

### Weaknesses and Questions
- I was hoping to see some broader discussion of metrics that could be used with such a methodology, since the paper is titled as being "metric-driven". The paper only discusses error as a metric, but I was curious to see how a methodology might adjust to other metrics like latency or memory consumption.
- The novelty of the approach as presented does not seem clear. As an example, there was no discussion or comparison with automatic quantization methods. PyTorch has [a tutorial on quantizing BERT to int8](https://pytorch.org/tutorials/intermediate/dynamic_quantization_bert_tutorial.html), but they do not discuss 8-bit float representations like this work. A broader consdieration about the space of training and testing quantized models would help strengthen the novelty of this work.
- The conclusion claims: "The search space for bit width allocation increases exponentially with more layers and more numerical
formats. " -- this seems true to me, but the paper didn't really present any evidence to highlight this and it only considered bit width allocation across the full network. The work would feel more strongly motivated if it included some analysis of how this search space grows if BERT, for instance, explored quantizing different layers with different formats

**Reviewer Expertise:**

Knowledgeable: I used to work in this area and/or I try to keep up with the literature but might not know the latest developments.